# Caffeine Inhibits NLRP3 Inflammasome Activation by Downregulating TLR4/MAPK/NF-κB Signaling Pathway in an Experimental NASH Model

**DOI:** 10.3390/ijms23179954

**Published:** 2022-09-01

**Authors:** Eduardo E. Vargas-Pozada, Erika Ramos-Tovar, Juan D. Rodriguez-Callejas, Irina Cardoso-Lezama, Silvia Galindo-Gómez, Daniel Talamás-Lara, Verónica Rocío Vásquez-Garzón, Jaime Arellanes-Robledo, Víctor Tsutsumi, Saúl Villa-Treviño, Pablo Muriel

**Affiliations:** 1Laboratory of Experimental Hepatology, Department of Pharmacology, Cinvestav-IPN, Mexico City 07360, Mexico; 2Postgraduate Studies and Research Section, School of Higher Education in Medicine-IPN, Plan de San Luis y Díaz Mirón s/n, Casco de Santo Tomás, Mexico City 11340, Mexico; 3Laboratory of Neuroplasticity and Neurodegeneration, Department of Pharmacology, Cinvestav-IPN, Mexico City 07360, Mexico; 4Department of Infectomics and Molecular Pathogenesis, Cinvestav-IPN, Mexico City 07360, Mexico; 5Laboratory of Fibrosis and Cancer, Faculty of Medicine and Surgery, ‘Benito Juárez’ Autonomous University of Oaxaca, UABJO, Oaxaca 68020, Mexico; 6Directorate of Catedras, National Council of Science and Technology CONACYT, Mexico City 03940, Mexico; 7Laboratory of Liver Diseases, National Institute of Genomic Medicine, INMEGEN, Mexico City 14610, Mexico; 8Department of Cell Biology, Cinvestav-IPN, Mexico City 07360, Mexico

**Keywords:** caffeine, non-alcoholic steatohepatitis, nucleotide-binding domain, leucine-rich-containing family, pyrin domain-containing-3 inflammasome, toll-like receptor 4, mitogen-activated protein kinase, nuclear factor-κB

## Abstract

Caffeine elicits protective effects against liver diseases, such as NASH; however, its mechanism of action involving the pyrin domain-containing-3 (NLRP3) inflammasome signaling pathway remains to be elucidated. This study aimed to evaluate the effect of caffeine on the NLRP3 inflammasome signaling pathway in a rat model of NASH. NASH was induced by feeding rats a high-fat, -sucrose, and -cholesterol diet (HFSCD) for 15 weeks along with a weekly low dose (400 mg/kg, i.p.) of CCl_4_. Caffeine was administered at 50 mg/kg p.o. The effects of HFSCD+CCl_4_ and caffeine on the liver were evaluated using biochemical, ultrastructural, histological, and molecular biological approaches. The HFSCD+CCl_4_-treated rats showed fat accumulation in the liver, elevated levels of inflammatory mediators, NLRP3 inflammasome activation, antioxidant dysregulation, and liver fibrosis. Caffeine reduced necrosis, cholestasis, oxidative stress, and fibrosis. Caffeine exhibited anti-inflammatory effects by attenuating NLRP3 inflammasome activation. Moreover, caffeine prevented increases in toll-like receptor 4 (TLR4) and nuclear factor-κB (NF-κB) protein levels and mitigated the phosphorylation of mitogen-activated protein kinase (MAPK). Importantly, caffeine prevented the activation of hepatic stellate cells. This study is the first to report that caffeine ameliorates NASH by inhibiting NLRP3 inflammasome activation through the suppression of the TLR4/MAPK/NF-κB signaling pathway.

## 1. Introduction

Non-alcoholic steatohepatitis (NASH) is a common liver disease that is primarily caused by the consumption of high-fat, cholesterol, and sugar diets and a sedentary lifestyle [1,2]. NASH is characterized by steatosis, hepatocyte injury, and inflammation, with or without fibrosis [3]. Neglected NASH can progress to cirrhosis, hepatocellular carcinoma, liver failure, or death [3,4].

Fat accumulation triggers reactive oxygen species (ROS) production in the liver and consequently induces necrosis and apoptosis [5]. Injured liver cells can be phagocytosed by Kupffer cells, which produce proinflammatory and profibrogenic cytokines, such as interleukins (ILs) and tumor necrosis factor-alpha (TNF-α) [6,7]. Proinflammatory cytokines stimulate nuclear factor-κB (NF-κB) activation, which triggers inflammasome activation. Inflammasomes are protein complexes that detect intracellular danger signals, and the most studied inflammasome is the nucleotide-binding domain, leucine-rich-containing family, pyrin domain-containing-3 (NLRP3) [8,9].

NF-κB promotes the synthesis of NLRP3, pro-IL-1β, and other proinflammatory cytokines. Cholesterol crystals, ROS, and fatty acids may activate inflammasomes, which function as damage-associated molecular patterns (DAMPs) [9,10]. These patterns bind and activate toll-like receptors (TLRs), inducing the activation of the NF-κB and mitogen-activated protein kinase (MAPK) signaling pathways, which are involved in inflammatory and fibrogenic responses [8,9,10].

When the inflammasome is activated, NLRP3, the effector pro-caspase-1, and the adapter protein apoptosis-associated speck-like protein containing a caspase recruitment domain (ASC) form a complex that leads to the activation of caspase-1 and the maturation of IL-1β, a cytokine that contributes to the maintenance of inflammatory and fibrogenic responses [8,9,10]. Inflammasomes are key proinflammatory regulators because they play essential roles in the initial metabolic stress, subsequent death of hepatocytes, and stimulation of fibrogenesis in NASH [11]. Blocking inflammasome activation mitigates hepatic inflammation and fibrosis in mice bearing NASH [8], suggesting that inhibition of NLRP3 inflammasome activation is a suitable pharmacological strategy against NASH.

Caffeine is a natural compound with beneficial effects on the liver [12,13]. Caffeine administration to experimental animals ameliorates liver scar tissue deposition, and frequent coffee consumption may reduce hepatic injury in patients with NASH [14,15,16,17,18,19]. Thus, caffeine may exert a therapeutic potential against NASH. However, the mechanism by which caffeine exerts its beneficial effects remains to be elucidated. Therefore, this investigation aimed to determine the effects of caffeine on the NLRP3 inflammasome and TLR4/MAPK/NF-κB signaling pathway in a rat model of NASH induced by an atherogenic diet and low CCl_4_ doses.

## 2. Results

### 2.1. Caffeine Prevents Liver Damage and Oxidative Stress in the NASH Experimental Model

Figure 1A shows representative macroscopic and microscopic images of hepatic tissues from the rats in the experimental groups. The specimens from the rats in the control and caffeine groups showed normal reddish color and did not present visible macroscopic or microscopic alterations, whereas those from the rats in the HFSCD+CCl_4_ and HFSCD+CCl_4_+caffeine groups turned yellowish-white (Figure 1A). Microscopically, the specimens from the rats in the HFSCD+CCl_4_ and HFSCD+CCl_4_+caffeine groups presented remarkable parenchymal disruption with enlarged hepatocytes, probably due to an exacerbated deposition of lipids and nuclear displacement toward the periphery of the cells. Interestingly, the specimens from the rats in the HFSCD+CCl_4_ group showed slight bands of connective tissue between hepatocytes, but this result was not observed in the specimens from the rats in the HFSCD+CCl_4_+caffeine group (Figure 1A).

The histologic alterations in the rats from the HFSCD+CCl_4_ group were accompanied by significant increases in ALT, γ-GTP, and AP serum activities (Figure 1B), but these increases in enzyme activities were attenuated in the tissues from the rats in the HFSCD+CCl_4_+caffeine group. The glycogen content in the tissues from the rats in the HFSCD+CCl_4_ group was almost depleted, and caffeine failed to prevent this effect (Figure 1B). In the tissues from the rats in the HFSCD+CCl_4_ group, LPO was increased and the GSH content was decreased, which induced oxidative stress, but the LPO and GSH alterations were significantly attenuated in the tissues from the rats in the HFSCD+CCl_4_+caffeine group (Figure 1C). Moreover, the tissues from the rats in the HFSCD+CCl_4_ group showed significantly increased 4-HNE and reduced Nrf2 levels, but these alterations were reversed in the tissues from the rats in the HFSCD+CCl_4_+caffeine group (Figure 1C). These parameters were not affected in the tissues from the rats in the caffeine group.

### 2.2. Effect of Caffeine on Liver Steatosis in the NASH Experimental Model

The rats in the HFSCD+CCl_4_ and HFSCD+CCl_4_+caffeine groups showed remarkable hepatic steatosis, as corroborated by the quantification of ORO staining, triglycerides, and cholesterol deposition in the liver (Figure 2A–C).

Moreover, the hepatic ultrastructure was disrupted in the HFSCD+CCl_4_ and HFSCD+CCl_4_+caffeine groups. The hepatic tissue contained abundant lipid droplets and disorganized mitochondria. Notably, transmission electron micrographs revealed that the hepatic parenchyma of the rats from the HFSCD+CCl_4_ group showed areas with collagen deposition, whereas that of the rats from the HFSCD+CCl_4_+caffeine group lacked collagen fibers (Figure 2B). The hepatic tissues of the rats from the caffeine group did not show significant changes in histology and ultrastructure, triglyceride content, or cholesterol content (Figure 2A). Western blot analysis revealed no significant differences in SREBP-1C and PPAR-α levels among the experimental groups (Figure 2D).

### 2.3. Caffeine Effect on the Proinflammatory and Fibrogenic TLR4-MAPK Pathway

TLR4 levels were significantly elevated in the hepatic tissues of the rats from the HFSCD+CCl_4_ group, but these results were prevented in the tissues of the rats from the HFSCD+CCl_4_+caffeine group (Figure 3A).

The JNK, ERK, and p38 protein levels in the hepatic tissue were not altered in any of the groups (Figure 3B–D). However, the levels of their phosphorylated forms were significantly elevated in the hepatic tissues from the rats in the HFSCD+CCl_4_ group, and these elevations were prevented in the HFSCD+CCl_4_+caffeine group (Figure 3E–G). The p-ERK/ERK, p-JNK/JNK, and p-p38/p38 ratios were elevated in the tissues from the rats in the HFSCD+CCl_4_ group, but these elevations were prevented in the HFSCD+CCl_4_+caffeine group (Figure 3H–J). No significant changes in these parameters were observed in the caffeine group.

### 2.4. Caffeine Prevents NASH by Blocking the Activation of the NLRP3 Inflammasome Signaling Pathway

Figure 4 shows the results of p65-specific antigen detection through IHC and western blot analyses. The percentages of tissue-positive areas and protein levels of p65 are shown in Figure 4B,E.

The hepatic tissues of the rats from the HFSCD+CCl_4_ group showed significantly increased p65 levels, but this effect was prevented in the HFSCD+CCl_4_+caffeine group. In addition, the protein expression levels of IL-17, IL-1β, and TNF-α were significantly higher in the hepatic tissues from the rats in the HFSCD+CCl_4_ group than in those from the rats in the HFSCD+CCl_4_+caffeine group (Figure 4F–H). Figure 4A shows the detection of NLRP3 and caspase 1 through IHC analysis. The percentages of positive areas of these proteins are shown in Figure 4C,D, respectively. The expression levels of these proteins increased in the tissues from the rats in the HFSCD+CCl_4_ group, but these increments were avoided in the HFSCD+CCl_4_+caffeine group. We evaluated the activation of the NLRP3 inflammasome through IF analysis (Figure 5A).

The colocalization between NLRP3 and caspase 1 proteins was clearly observed in the tissues from the rats in the HFSCD+CCl_4_ group, indicating the formation of the active NLRP3 inflammasome complex. Quantification of fluorescence intensity of these proteins corroborated this observation (Figure 5B). Notably, this phenomenon was prevented in the HFSCD+CCl_4_+caffeine group, and no alteration was observed in the caffeine group. Western blot analysis results showed that the protein expression levels of NLRP3, ASC, and caspase 1 in the NLRP3 inflammasome complex were significantly increased in the hepatic tissues of the rats from the HFSCD+CCl_4_ group (Figure 5C), but this phenomenon was prevented in the HFSCD+CCl_4_+caffeine group. 

### 2.5. Caffeine Attenuates Fibrosis by Mitigating Hepatic Stellate Cell Activation

IF analysis revealed that the protein expression levels of NLRP3 and α-SMA were increased and colocalized in the tissues of the rats from the HFSCD+CCl_4_ group (Figure 6A), and the quantification validated this observation, suggesting a close relationship between the levels of both proteins and indicating the activation of hepatic stellate cells (HSC) [4]. Notably, the expression levels of these proteins were decreased in the tissues of the rats from the HFSCD+CCl_4_+caffeine group (Figure 6B). 

Moreover, ECM accumulation was evidenced via Masson’s trichrome staining and quantified by assessing liver hydroxyproline content (Figure 7A–C). ECM deposition was increased in the HFSCD+CCl_4_ group, but this accumulation was attenuated in the HFSCD+CCl_4_+caffeine group. No significant effects were observed in the caffeine control group. Scar tissue turnover was assessed by determining the activities of MMP-2 and MMP-9, and MMP-13 protein levels (Figure 7D–G). The activities of MMP-9 and MMP-2 were significantly higher in the hepatic tissues from the rats in the HFSCD+CCl_4_+caffeine group than in those from the rats in the HFSCD+CCl_4_ group (Figure 7F,G). By contrast, the protein level of MMP-13 was significantly increased in the hepatic tissues from the rats in the HFSCD+CCl_4_ group, and this alteration was prevented in the hepatic tissues from the rats in the HFSCD+CCl_4_+caffeine group (Figure 7E).

IHC and western blot analysis results showed that the protein expression levels of TGF-β, α-SMA, CTGF, and desmin were significantly increased in the hepatic tissues from the rats in the HFSCD+CCl_4_ group, but this effect was significantly prevented in the hepatic tissues from the rats in the HFSCD+CCl_4_+caffeine group (Figure 8A–G). Smad7 levels were reduced in the hepatic tissues from the rats in the HFSCD+CCl_4_ and HFSCD+CCl_4_+caffeine groups (Figure 8H). No alterations in these protein levels were observed in the hepatic tissues from the rats in the caffeine group.

## 3. Discussion

NASH is a common liver disease worldwide without effective pharmacological treatment. It is a chronic and progressive liver disease that, in addition to steatosis, induces an inflammatory process that can promote fibrosis, cirrhosis, and hepatocellular carcinoma [1,2]. The NLRP3 inflammasome plays a pivotal role in NASH progression [8], making it an attractive target for the treatment of NASH. Caffeine has been shown to protect the liver from acute and chronic damage of diverse etiologies through antioxidant, anti-inflammatory, antifibrotic, and antisteatotic mechanisms [8,9,12]. However, the role of caffeine in the NLRP3 inflammasome signaling pathway had not yet been investigated. In the present study, we report for the first time that caffeine mitigates experimental NASH by inhibiting NLRP3 inflammasome activation and decreasing the TLR4/MAPK/NF-κB pathway. We also provide evidence that caffeine prevents oxidative stress, inflammation, and fibrogenesis by improving the Nrf2 signaling pathway and decreasing HSC activation in NASH (Figure 9).

Studies show that regular coffee consumption (1 to 3 cups of coffee containing 40 to 180 mg of caffeine each) results in plasma concentrations of 0.5 to 2.6 µg/mL of caffeine in humans [20,21,22]. Moreover, a study in male Sprague–Dawley rats showed that a daily caffeine intake of 38.7 to 51.1 mg/kg for 12 weeks led to a caffeine plasma concentration of 1.62 ± 0.53 µg/mL [22]. Based on these results and considering that we administered 50 mg/kg per day caffeine, the caffeine plasma concentrations were probably within a range of 1.62 ± 0.53 µg/mL, which is equivalent to the consumption of one to three cups of coffee daily in humans. In addition, we have used this dose in previous research, and we found no adverse effects [23].

It is known that the development of NAFLD and NASH is closely related to metabolic syndrome, which includes disorders such as high blood pressure, high blood sugar, excess body fat around the waist, abnormal blood lipid levels, and alteration in the gut micro-biome. In this sense, previous investigations have shown that no associations were found between caffeine concentrations with total cholesterol and low-density lipoprotein levels either in caffeine drug users or nonusers [24], while another report showed that caffeine does not interfere with the lipid profile in cyclists [25]. Concerning the intestinal microbiome, a study showed that caffeine is directly associated with changes only in some intestinal microbiota groups, such as the Bacteroides group [26]. Another study showed that higher caffeine consumption was associated with increased richness and evenness of the mucosa-associated gut microbiota, and higher relative abundance of anti-inflammatory bacteria, such as Faecalibacterium and Roseburia, and lower levels of potentially harmful Erysipelatoclostridium [27]. Regarding other metabolic disorders, it is known that in persons who have not previously consumed caffeine, caffeine intake raises blood pressure in the short term, and affect tolerance develops within a week but may be incomplete in some person results in a modest increase in systolic and diastolic blood pressure. Experimental studies in humans do not show an association between caffeine intake and atrial fibrillation. Other findings indicate that the consumption of caffeinated coffee is not associated with an increased risk of cardiovascular events in the general population or among persons with a history of hypertension, diabetes, or cardiovascular diseases. Metabolic studies suggest that caffeine may improve energy balance by reducing appetite and increasing the basal metabolic rate and food-induced thermogenesis. Limited evidence from randomized trials also supports a modest beneficial effect of caffeine intake on body fatness. Consumption of caffeinated coffee for up to 6 months does not affect insulin resistance [28]. 

Our results are consistent with previous findings that caffeine protects against liver damage [23,29,30,31,32,33,34]. We also found that caffeine does not prevent liver lipid accumulation, which suggests that its hepatoprotective effects are not due to its antisteatotic capability. By contrast, other researchers have reported that caffeine exerts antisteatotic effects. These opposing effects may be ascribed to the fact that the effect of caffeine was determined in combination with other compounds [17,18]. According to our results, caffeine does not show any improvement in hepatocyte steatosis but seems to protect against NASH via its anti-inflammation effect.

Oxidative stress plays an important role in the progression of liver damage during NASH pathology [5,32]. In the present study, caffeine protected against oxidative stress by mitigating LPO products, such as MDA and 4-HNE. This effect can be attributed to the free radical scavenging capability of caffeine, which has a high affinity for free radicals and neutralizes them by donating electrons and reducing their adverse reactivity [35,36]. Additionally, caffeine attenuates oxidative stress by activating the Nrf2 signaling pathway [30]. In the present study, caffeine preserved normal hepatic Nrf2 levels in the HFSCD+CCl_4_-treated rats. Notably, attenuation of oxidative stress may decrease inflammation and fibrosis induction [32]. Our results showed that caffeine effectively mitigated oxidative stress markers, inflammation, and fibrosis in the NASH model.

The NLRP3 inflammasome signaling pathway plays a crucial role in liver diseases [37,38]. Evidence indicates that IL-1β maturation via the NLRP3 inflammasome pathway correlates with the progression of inflammation and fibrosis in NASH [8], and patients with this disease show increased activation of this pathway [37,38,39]. Therefore, we investigated whether caffeine exerts its beneficial effects by inhibiting NLRP3 inflammasome activation in our NASH model. IF analysis results showed increased colocalization of inflammasome-associated proteins, namely, NLRP3 and caspase 1, suggesting that they interacted to form the active NLRP3 inflammasome complex. The present study is the first to report that caffeine prevents the activation of the NLRP3 inflammasome and the increases in NLRP3, ASC, caspase 1, and IL-1β protein levels, which are consistent with results obtained from in vitro models [40,41], a model of lung damage [42], and experimental models of neuroinflammation [43,44]. NLRP3 inflammasome is most prominently expressed in Kupffer cells and liver sinusoidal endothelial cells and moderately expressed in periportal myofibroblasts, hepatic stellate cells, and hepatocytes [10,45]; therefore, caffeine NLRP3 inhibition may principally occur in Kupffer cells. On the other hand, free fatty acids that originate from lipolysis of triglyceride in adipose tissue are delivered through the blood to the liver. The other major contributor to the free fatty acid flux through the liver is de novo lipid synthesis, the process by which hepatocytes convert excess carbohydrates, especially fructose, to fatty acids. The two major fates of fatty acids in hepatocytes are mitochondrial beta-oxidation and re-esterification to form triglyceride. Triglyceride can be exported into the blood as very low-density lipoprotein or stored in lipid droplets. Lipid droplet triglyceride undergoes regulated lipolysis to release fatty acids back into the hepatocyte-free fatty acid pool [11,46]. Our results show no significant effect of caffeine on the fatty liver but an important anti-inflammatory effect; therefore, it seems that caffeine exhibits a cell-specific effect, acting on proinflammatory cells but not on hepatocytes. However, the effect of caffeine on other cell types cannot be completely discarded with the present data. Thus, our findings contribute to illuminating the molecular mechanisms associated with the beneficial effects of caffeine in treating liver diseases. The present study may serve as a basis for understanding NASH pathology and identifying therapeutic targets.

Previous studies have shown that DAMPs, such as ROS, trigger NF-κB and MAPK activation to upregulate IL-1β [9,10,40]. Additionally, activated NF-κB translocates to the nucleus to upregulate the expression of NLRP3, IL-1β, and other inflammatory and fibrogenic cytokines [8,9,40]. Activated NLRP3 inflammasome and IL-1β promote the production of the proinflammatory and profibrogenic IL-17, which contributes to ECM-exacerbated deposition within the hepatic parenchyma [10,45,47]. In the present study, caffeine markedly prevented the increase in NF-κB protein levels induced by HFSCD+CCl_4_ treatment. On the other hand, TLR4, a receptor that stimulates several downstream pathways, including the MAPK and NF-κB pathways [48], acts as a key regulator of the NLRP3 inflammasome pathway [49]. The present results showed that caffeine prevented the increase in the TLR4 protein level and significantly impaired NLRP3 inflammasome activation in the HFSCD+CCl_4_-treated rats. The MAPK signaling pathway is also involved in the activation of the NF-κB pathway and upregulation of IL-1β [40,50,51]. In the present study, caffeine prevented HFSCD+CCl_4_-induced JNK, ERK, and p38 phosphorylation. These results are consistent with those of previous studies using different experimental models and tissues [52,53,54]. Furthermore, caffeine prevents the increase in TGF-β, an efficient activator of the profibrogenic HSCs and the main inducer of ECM-exacerbated deposition [4,7], thereby inhibiting α-SMA protein elevation, an indicator of HSC activation. Therefore, pharmacological inhibition of TGF-β by caffeine reduced fibrosis induced by HFSCD+CCl_4_ treatment. Taken together, our findings indicate that caffeine prevents NLRP3 inflammasome activation and inhibits IL-1β upregulation by interfering with TLR4 receptor-mediated NF-κB and MAPK activities, resulting in the suppression of inflammation and fibrosis during NASH progression. Our results provide new evidence for the therapeutic potential of caffeine in the treatment of chronic liver diseases, particularly NASH alongside fibrosis. The main limitation of this study is that caffeine was administered along with HFSCD and CCl_4_. Therefore, further studies need to be done to investigate the ability of caffeine to reverse experimental NASH. Although our investigation shows clear evidence of the role of caffeine in preventing NASH-associated alterations, additional preclinical and clinical studies are warranted before caffeine can be prescribed as a treatment for NASH.

## 4. Materials and Methods

### 4.1. Animal Treatments 

#### Non-Alcoholic Steatohepatitis Induction and Caffeine Treatment

Rats were obtained from the animal production and experimentation unit of the Center for Research and Advanced Studies of the National Polytechnic Institute (UPEAL–CINVESTAV–IPN; Mexico City, Mexico). Thirty-two Wistar male rats weighing 100–120 g were fed a high-fat, -sucrose, and -cholesterol diet (HFSCD) ad libitum [55] and injected intraperitoneally with CCl_4_ at 400 mg/kg once a week to induce NASH. The rats were divided into four groups (n = 8 per group): control group, fed a standard diet ad libitum (Labdiet^®^ No. 5053, Indianapolis, IN, USA); HFSCD+CCl_4_ group, subjected to HFSCD plus CCl_4_ treatment; HFSCD+CCl_4_+caffeine group, subjected to HFSCD+CCl_4_ treatment plus caffeine at 50 mg/kg body weight daily via gavage; and caffeine group, subjected to caffeine alone. All treatments were administered for 15 weeks. The rats were housed in polycarbonate cages under controlled conditions (21 ± 1 °C, 50–60% relative humidity, and 12-h dark/light cycles) and water ad libitum. At the end of the experiment, the animals were anesthetized with ketamine and xylazine and euthanized by exsanguination. Blood was collected by heart puncture and then centrifuged at 12,000× *g* to obtain serum and hepatic samples, which were stored at −75 °C. The study complies with the Cinvestav’s guidelines, the Mexican official regulation (NOM-062-ZOO-1999), and technical specifications for the production, care, and handling of laboratory animals and follows the Guide for the Care and Use of Laboratory Animals (NRC, 2011). The Ethics Committee of the Animal Lab Facility of Cinvestav approved the study (protocol number 281-19).

### 4.2. Reagents

Caffeine, cholesterol, sucrose, and cholate were purchased from Sigma-Aldrich (St. Louis, MO, USA). CCl_4_ was obtained from J.T. Backer (Xalostoc, Mexico City, Mexico). Gloria^®^ brand unsalted butter (Cremería Americana S.A. de C.V., Mexico City, Mexico) and rennet casein powder 30 mesh size (Irish Dairy Board Proteins and Ingredients, Dublin, Ireland) were used in this study.

### 4.3. Antibodies

Table 1 shows the commercial information of antibodies against c-Jun N-terminal kinases (JNKs), extracellular signal-regulated kinases (ERKs), p38, p-JNK, p-ERK, p-p38, metalloproteinase (MMP)-13, Smad7, transforming growth factor-beta (TGF-β), smooth muscle alpha-actin (α-SMA), β-actin, desmin, connective tissue growth factor (CTGF), 4-hydroxynonenal (4-HNE), nuclear factor erythroid 2-related factor 2 (Nrf2), IL-1β, NF-κB-p65, TNF-α, TLR4, sterol regulatory element-binding protein 1C (SREBP1C), peroxisome proliferator-activated receptor alpha (PPAR-α), NLRP3, ASC, and caspase 1. The antibody dilutions used for immunofluorescence (IF), immunohistochemistry (IHC), and western blot assays were 1:250, 1:250, and 1:500, respectively.

### 4.4. Biochemical Analyses

Alanine aminotransferase (ALT) [56], gamma-glutamyl transpeptidase (γ-GTP) [57], and alkaline phosphatase (AP) [58] activities were measured in the rat plasma. Liver glycogen content and reduced glutathione (GSH) levels were quantified as described previously [59,60]. The level of lipid peroxidation in hepatic tissue was measured by quantifying the malondialdehyde (MDA) content using the thiobarbituric acid procedure [61]. Liver triglyceride levels were determined using a commercial kit (TR0100, Sigma-Aldrich^®^, St. Louis, MO, USA) as previously described [62]. Liver cholesterol levels were determined as previously described [63]. Protein concentration was determined using the Bradford method, and BSA was used as the standard [64]. The liver extracellular matrix (ECM) was assessed in fresh tissue samples using Ehrlich’s reagent (dimethylaminobenzaldehyde) as previously described [65].

### 4.5. Histology Determinations

Hepatic tissue was fixed in 4% paraformaldehyde for 48 h, dehydrated, embedded in paraffin, and then cut into 5-µm-thick sections. The sections were deparaffinized in xylene for 45 min, rehydrated in a graded alcohol series, and then prepared for Masson trichrome, red oil O (ORO), hematoxylin and eosin (H&E) staining, or IHC analysis. The sections were sealed with neutral resin, and images were captured under a light microscope (80i, Eclipse, Nikon^®^, Tokyo, Japan).

### 4.6. Immunohistochemistry Assays

Hepatic sections were dewaxed, hydrated with xylene and alcohol, immersed in PBS, autoclaved with 0.10 N citrate buffer, and then washed again with PBS. Endogenous peroxidase was blocked with methanol peroxidase, and the samples were washed with PBS. Then, 5% skim milk in PBS was used to block nonspecific binding. The sections were incubated overnight with primary antibodies (Table 1) diluted in 3% fetal bovine serum, rinsed with PBS, and then incubated with secondary antibodies (Table 1). The stained specimens were covered with resin. Images were captured with an 80i Eclipse microscope (Nikon^®^, Tokyo, Japan) [66], and positive signals (brown gradients) were quantified using the ImageJ^®^ software (version 1.53q; National Institutes of Health, Bethesda, MD, USA) [67].

### 4.7. Transmission Electron Microscopy

Fresh liver samples were fixed with 2.5% (*v*/*v*) glutaraldehyde in 0.1 M sodium cacodylate buffer (pH 7.2) and then washed with 0.1 M sodium cacodylate. The samples were post-fixed with 2% (*w*/*v*) osmium tetroxide in 0.2 M sodium cacodylate, washed with 0.1 M sodium cacodylate, dehydrated using increasing concentrations of ethanol, and then embedded in epoxy resin. Semi-thin sections were stained with toluidine blue. Thin sections were contrasted with uranyl acetate and lead citrate and examined under a JEOL JEM-1011 transmission electron microscope (JEOL, Ltd., Tokyo, Japan) [68].

### 4.8. Immunofluorescence

The tissue sections were subjected to double-labeling IF of NLRP3/caspase 1 and NLRP3/α-SMA as previously described [69] and then permeabilized with 0.2% PBS-Triton. Thereafter, the sections were treated with 5% BSA, incubated overnight at 4 °C with primary antibodies (Table 1), washed with 0.2% PBS-triton, and then incubated with secondary antibodies (Table 1) diluted in 0.2% PBS-triton. Control sections were processed without primary antibodies. All sections were co-incubated with 4′, 6′-diamidino-2 phenylindole (Invitrogen, Waltham, MA, USA, 1:1000) in 0.2% PBS-triton, washed, and then mounted on glass slides. Dry sections were covered and slipped with VectaShield mounting medium (Vector Laboratories, Newark, CA, USA).

### 4.9. Western Blotting Analysis 

Protein analysis via western blotting was performed as previously described [66]. The protein concentration was measured using the bicinchoninic acid method [70]. The primary antibodies used are listed in Table 1. β-Actin was used as a loading control, and the band intensity was captured in photographic plates (Cat. number 822526, Kodak^®^, New York, NY, USA) and then measured through densitometric scanning using the ImageJ^®^ software (version 1.53q; National Institutes of Health, Bethesda, MD, USA) [67].

### 4.10. Zymography

The activity of MMPs was assessed on gelatin-substrate gels as previously described (33). Digitalized images were used to quantify band intensity using the ImageJ^®^ software (version 1.53q; National Institutes of Health, Bethesda, MD, USA) [67].

### 4.11. Statistical Analyses

Results are expressed as the mean ± standard error of the mean (SEM). The GraphPad by Dotmatics Prism^®^ software (version 7.0; San Diego, CA, USA) and one-way ANOVA were used to perform multiple comparisons and compare multiple groups, respectively. Tukey’s test was performed, and statistical significance was considered at *p* ≤ 0.05.

## 5. Conclusions

Cumulative evidence indicates that caffeine exerts protective effects against liver disease. However, to the best of our knowledge, the mechanism by which caffeine, as an anti-inflammatory drug, targets the NLRP3 inflammasome in NASH remains to be elucidated. Results showed that caffeine significantly reduced the NLRP3 protein level, NLRP3 inflammasome activation, and TLR4/MAPK/NF-κB signaling pathways in a rat model of NASH induced by an atherogenic diet and low CCl_4_ doses. Thus, this study describes a novel mechanism by which caffeine exerts its hepatoprotective effects against NASH and supports previous suggestions that this alkaloid is effective in treating human NASH.

## Figures and Tables

**Figure 1 ijms-23-09954-f001:**
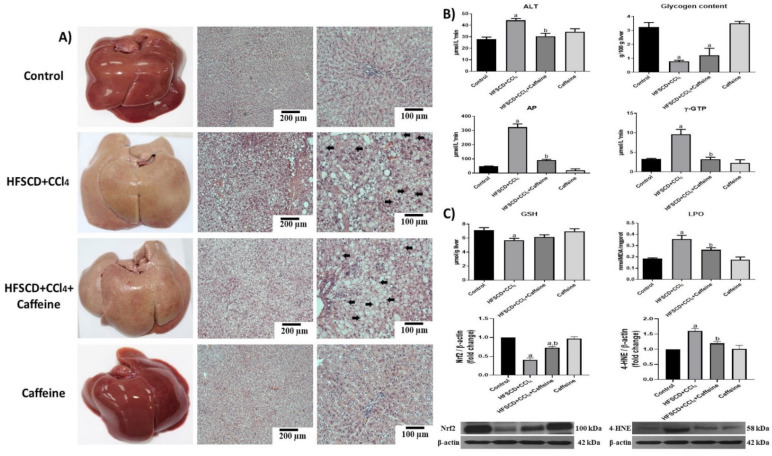
Effect of caffeine on the liver structure, liver damage, and oxidative stress markers in rats fed a high-fat, -sucrose, and -cholesterol diet plus CCl_4_ (HFSCD+CCl_4_ protocol or NASH model). (**A**) Macroscopic anatomy and microscopic liver structure. H&E staining of hepatic tissues from rats in the control, HFSCD+CCl_4_, HFSCD+CCl_4_+caffeine, and caffeine groups. Scale bar = 200 and 100 μm. Arrows show nuclei displaced toward the periphery of the cells. (**B**) Markers of liver damage. Serum alanine aminotransferase (ALT), gamma-glutamyl transpeptidase (γ-GTP), alkaline phosphatase (AP) activities, and hepatic glycogen content. Bars represent the mean of experiments performed in duplicate ± SEM (n = 8). (**C**) Oxidative stress markers. Degree of lipid peroxidation (LPO) and content of reduced glutathione (GSH). Bars represent the mean of experiments performed in duplicate ± SEM (n = 8). Western blot of 4-hydroxynonenal (4-HNE) and nuclear factor-E2-related factor 2 (Nrf2) proteins. Bars represent the mean of experiments performed in duplicate ± SEM (n = 3): β-actin was used as a loading control. (a) *p* < 0.05 compared with the control group; (b) *p* < 0.05 compared with the HFSCD+CCl_4_ group.

**Figure 2 ijms-23-09954-f002:**
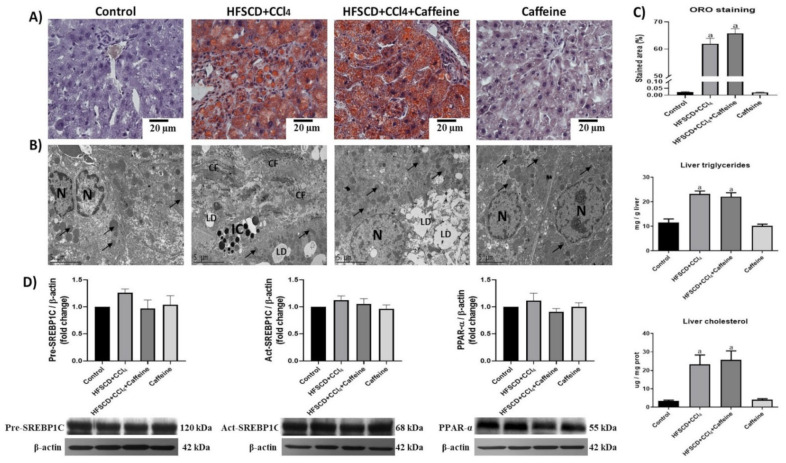
Effect of caffeine on steatosis markers and ultrastructure anatomy of hepatic tissue from the NASH model. (**A**) ORO staining of hepatic tissues from rats in the control, HFSCD+CCl_4_, HFSCD+CCl_4_+caffeine, and caffeine groups. Scale bar = 20 μm. (**B**) Transmission electron micrographs of hepatic sections. Scale bar = 2 μm. (Arrow) mitochondria, (CF) collagen fibers, (IC) inflammatory cell, (LD) lipid drop, (N) nucleus. (**C**) Percentage of labeled area stained with ORO. Bars represent the mean of experiments performed in duplicate ± SEM (n = 4), liver triglycerides, and cholesterol. Bars represent the mean of experiments performed in duplicate ± SEM (n = 5). (**D**) Western blot analysis of precursor-sterol regulatory element-binding protein 1C (pre-SREBP1C), active-SREBP1C (act-SREBP1C), and peroxisome proliferator-activated receptors alpha (PPAR-α) proteins. Bars represent the mean of experiments performed in duplicate ± SEM (n = 3); β-actin was used as a loading control. (a) *p* < 0.05 compared with the control group.

**Figure 3 ijms-23-09954-f003:**
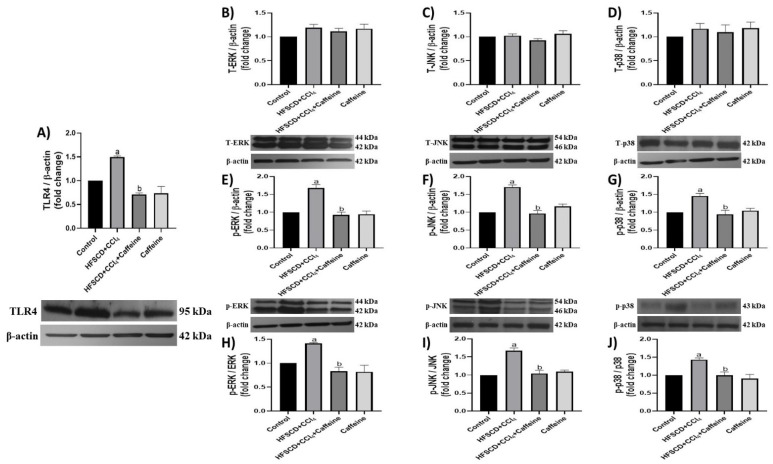
Effect of caffeine on the TLR4/MAPK pathway of hepatic tissue from the NASH model. Protein levels of TLR4 (**A**), ERK (**B**), JNK (**C**), p38 (**D**), p-ERK (**E**), p-JNK (**F**), and p-p38 (**G**) in the hepatic tissues from rats in the control, HFSCD+CCl_4_, HFSCD+CCl_4_+caffeine, and caffeine groups were determined using western blot analysis. Bars represent the mean of experiments performed in duplicate ± SEM (n = 3); β-actin was used as a loading control. p-ERK/ERK (**H**), p-JNK/JNK (**I**), and p-p38/p38 (**J**) ratios. Values are shown as a fold increment in optical density normalized to control group values (control = 1). Bars represent the mean of experiments performed in duplicate ± SEM. (a) *p* < 0.05 compared with the control group; (b) *p* < 0.05 compared with the HFSCD+CCl_4_ group.

**Figure 4 ijms-23-09954-f004:**
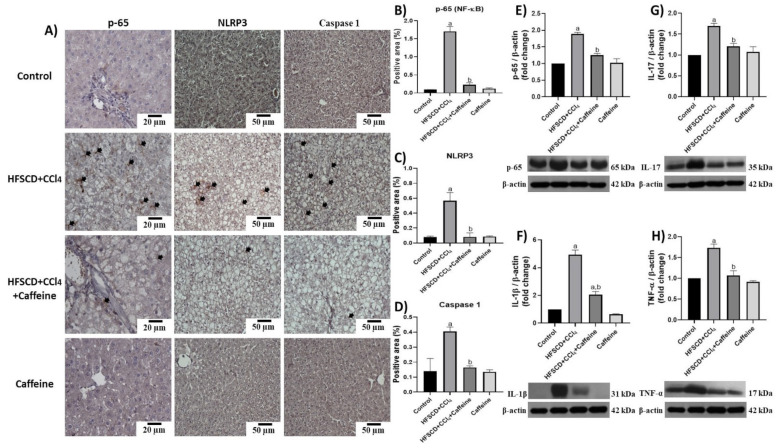
Effect of caffeine on NF-κB (p65) level, proinflammatory cytokines, and NLRP3 of hepatic tissue from the NASH model. (**A**) Representative IHC images of p65, NLRP3, and caspase 1 of hepatic tissues from rats in the control, HFSCD+CCl_4_, HFSCD+CCl_4_+caffeine, and caffeine groups. Scale bar = 50 μm and 20 μm. Percentage of positive area for p65 (**B**), NLRP3 (**C**), and caspase 1 (**D**) obtained from IHC slices (n = 4). Arrows indicate a positive label. Levels of p65 (**E**), interleukin (IL)-1β (**F**), IL-17 (**G**), and TNF-α (**H**). Proteins in hepatic tissues were detected using western blot analysis (n = 3). β-Actin was used as a loading control. Values are presented as fold increments in optical density normalized to control group values (control = 1). Bars represent the mean ± SEM. (a) *p* < 0.05 compared with the control group; (b) *p* < 0.05 compared with the HFSCD+CCl_4_ group.

**Figure 5 ijms-23-09954-f005:**
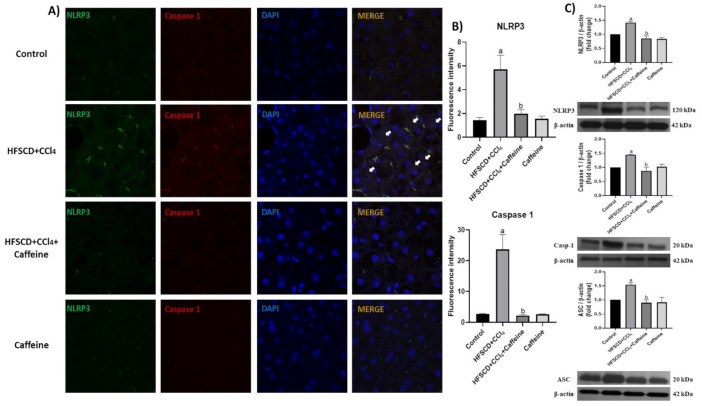
Effect of caffeine on NLRP3 inflammasome activation in hepatic tissue from the NASH model. (**A**) IF analysis of NLRP3/caspase 1 in hepatic tissues from rats in the control, HFSCD+CCl_4_, HFSCD+CCl_4_+caffeine, and caffeine groups. White arrow = Colocalization between NLRP3 inflammasome and pro-caspase-1 marks. (**B**) IF intensity quantification of NLRP3 and caspase-1 (n = 3). (**C**) Levels of NLRP3, caspase-1, and ASC proteins detected using western blot (n = 3); β-actin was used as a loading control. Values are shown as a fold increment in optical density normalized to control group values (control = 1). Bars represent the mean ± SEM. (a) *p* < 0.05 compared with the control group; (b) *p* < 0.05 compared with the HFSCD+CCl_4_ group. The slides were observed under a confocal microscope at 63× magnification.

**Figure 6 ijms-23-09954-f006:**
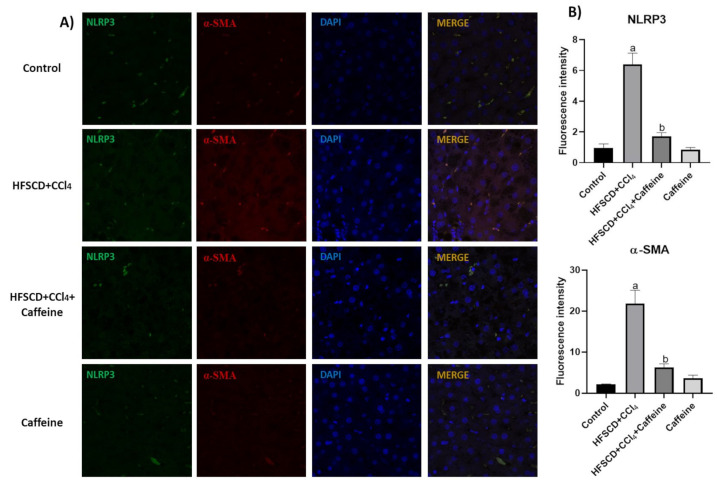
Effect of caffeine on NLRP3/α-SMA dual-labeling in hepatic tissue from the NASH model. (**A**) IF analysis of NLRP3/α-SMA in hepatic tissues from rats in the control, HFSCD+CCl_4_, HFSCD+CCl_4_+caffeine, and caffeine groups. (**B**) Quantification of fluorescence intensity of NLRP3 and α-SMA (n = 3). Bars represent the mean ± SEM. (a) *p* < 0.05 compared with the control group; (b) *p* < 0.05 compared with the HFSCD+CCl_4_ group. The slides were observed under a confocal microscope at 63× magnification.

**Figure 7 ijms-23-09954-f007:**
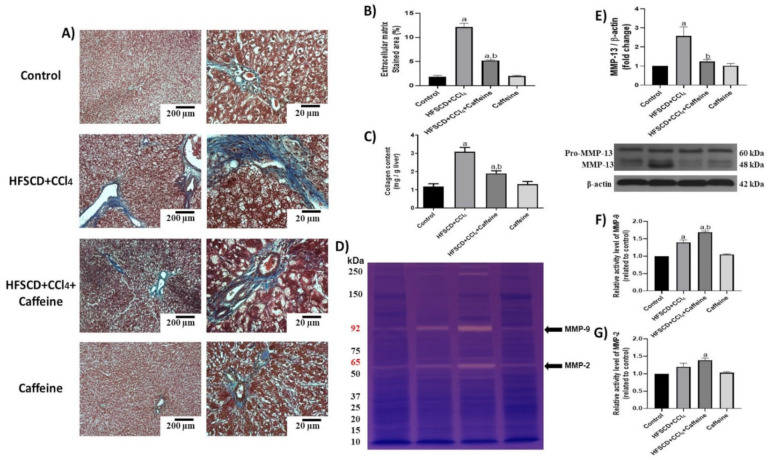
Effect of caffeine on fibrosis and profibrogenic mediators and MMPs in hepatic tissue from the NASH model. (**A**) Masson’s trichrome staining in hepatic tissues from rats in the control, HFSCD+CCl_4_, HFSCD+CCl_4_+caffeine, and caffeine groups. Scale bars = 200 and 20 μm. (**B**) Percentages of collagen areas (n = 4). (**C**) Collagen content was determined by measuring the liver hydroxyproline level (n = 6). Activities of MMP-9 (**F**) and MMP-2 (**G**) were determined using zymography (**D**) (n = 5). Bars represent the mean ± SEM. Values are shown as fold increments in optical density normalized to control group values (control = 1). (**E**) Protein level of MMP-13 was examined using western blot analysis (n = 3); β-actin was used as a loading control. Bars represent the mean ± SEM. (a) *p* < 0.05 compared with the control group; (b) *p* < 0.05 compared with the HFSCD+CCl_4_ group.

**Figure 8 ijms-23-09954-f008:**
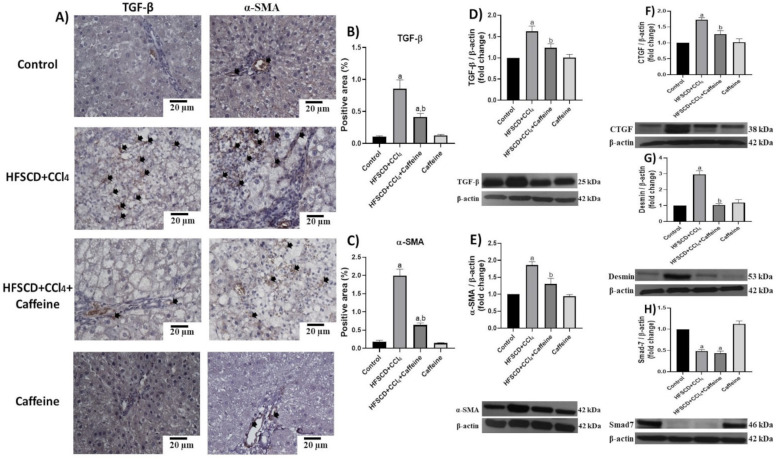
Caffeine maintains basal levels of TGF-β, α-SMA, and profibrogenic mediators in hepatic tissue from the NASH model. (**A**) Representative IHC images of TGF-β and α-SMA in the control, HFSCD+CCl_4_, HFSCD+CCl_4_+caffeine, and caffeine groups. Scale bar = 20 μm. Positive areas for TGF-β (**B**) and α-SMA (**C**) are shown in the histogram (n = 4). Protein levels of TGF-β (**D**), α-SMA (**E**), CTGF (**F**), desmin (**G**), and Smad7 (**H**) were detected using western blot analysis (n = 3); β-actin was used as a loading control. Values are shown as a fold increment in optical density normalized to control group values (control = 1). Bars represent the mean ± SEM. (a) *p* < 0.05 compared with the control group; (b) *p* < 0.05 compared with the HFSCD+CCl_4_ group.

**Figure 9 ijms-23-09954-f009:**
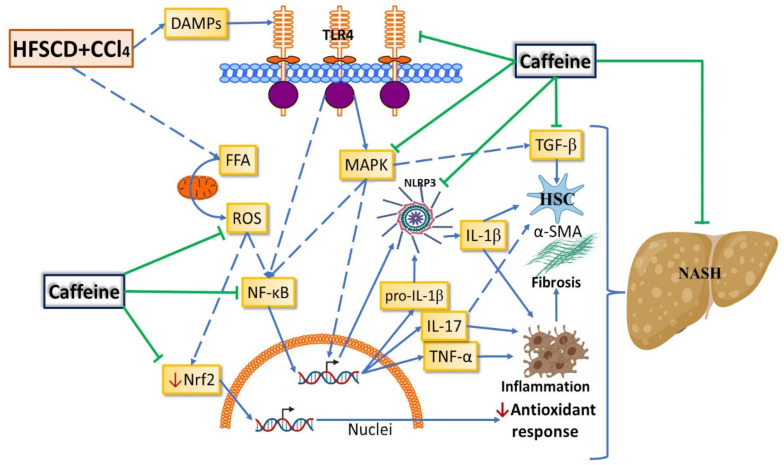
Schematic diagram showing how caffeine affects the NASH experimental model. Caffeine attenuates the experimental NASH progression possibly by modulating the nuclear factor erythroid 2-related factor 2 (Nrf2) and nuclear factor-κB (NF-κB) signaling pathways, thereby reducing oxidative stress and inflammation, respectively. Caffeine also attenuates fibrosis progression by modulating mitogen-activated protein kinases (MAPK) and transforming growth factor-beta (TGF-β) signaling pathways, thereby decreasing the activation of hepatic stellate cells (HSCs). Caffeine inhibits inflammasome activation by blocking the toll-like receptor 4 (TLR4)/MAPK/NF-κB signaling pathway. ROS: reactive oxygen species; DAMPs: damage-associated molecular patterns; α-SMA: smooth muscle alpha-actin; IL: interleukin; TNF-α: tumor necrosis factor-alpha.

**Table 1 ijms-23-09954-t001:** Antibodies used in western blot, immunohistochemistry and immunofluorescence techniques.

Protein	Brand (Location)	Catalog	WB Dilution	IHC Dilution	IF Dilution
MMP-13	Merck-Millipore^®^ (Burlington, MA, USA)	MAB13426	1:500		
TGF-β	Merck-Millipore^®^ (Burlington, MA, USA)	MAB1032	1:500	1:250	
NF-κB (p65)	Merck Millipore^®^ (Burlington, MA, USA)	MAB3026	1:500	1:250	
JNK	Cell Signaling Technology^®^ (Danvers, MA, USA)	9252S	1:500		
pp38	Cell Signaling Technology^®^ (Danvers, MA, USA)	9211S	1:500		
p38	Abcam^®^ (Cambridge, UK)	AB31828	1:500		
Smad7	Abcam^®^ (Cambridge, UK)	AB90086	1:500		
pJNK	Abcam^®^ (Cambridge, UK)	AB131499	1:500		
4-HNE	Abcam^®^ (Cambridge, UK)	AB46545	1:500		
IL-1β	Abcam^®^ (Cambridge, UK)	AB18329	1:500		
Nrf2	Abcam^®^ (Cambridge, UK)	AB31163	1:500		
SREBP1C	Abcam^®^ (Cambridge, UK)	AB28481	1:500		
α-SMA	Sigma-Aldrich^®^ (St. Louis, MI, USA)	A-5691	1:500	1:250	1:250
Desmin	Sigma-Aldrich^®^ (St. Louis, MI, USA)	243M-1	1:500		
pERK	Santa Cruz Biotechnology^®^ (Santa Cruz, CA, USA)	SC-136521	1:500		
ERK	Santa Cruz Biotechnology^®^ (Santa Cruz, CA, USA)	SC-292838	1:500		
CTGF	Santa Cruz Biotechnology^®^ (Santa Cruz, CA, USA)	SC-365970	1:500		
IL-17	Santa Cruz Biotechnology^®^ (Dallas, TX, USA)	SC-374218	1:500		
TNF-α	Santa Cruz Biotechnology^®^ (Dallas, TX, USA)	SC-52746	1:500		
ASC	Santa Cruz Biotechnology^®^ (Dallas, TX, USA)	SC-514414	1:500		
Caspase 1	Santa Cruz Biotechnology^®^ (Dallas, TX, USA)	SC-392736	1:500	1:250	1:250
PPAR-α	Santa Cruz Biotechnology^®^ (Dallas, TX, USA)	SC-398394	1:500		
NLRP3	Novus biologicals^®^ (Littleton, CO, USA)	NBP2-12446	1:500	1:250	1:250
TLR4	Thermo Fisher^®^ (Waltham, MA, USA)	48-2300	1:500		
β-actin	Thermo Fisher^®^ (Waltham, MA, USA)	AM4302	1:500		
**Secondary Antibodies**					
Anti-Rabbit	Merck Millipore^®^ (Burlington, MA, USA)	A0545	1:3000	1:1000	
Anti-Mouse	Thermo Fisher^®^ (Waltham, MA, USA)	62-6520	1:5000	1:1000	
Alexa fluor 594	Thermo Fisher^®^ (Waltham, MA, USA)	Z25007			1:1000
Alexa fluor 488	Thermo Fisher^®^ (Waltham, MA, USA)	Z25302			1:1000

WB: Western blotting; IHC: immunohistochemistry IF: immunofluorescence.

## Data Availability

The datasets generated and analyzed during the current study are available on reasonable request.

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
