# Peer review of "Caffeine Inhibits NLRP3 Inflammasome Activation by Downregulating TLR4/MAPK/NF-κB Signaling Pathway in an Experimental NASH Model"

_ijms, 2022, doi:10.3390/ijms23179954_

Round 1

Reviewer 1 Report

The authors test caffeine effect in a NASH model, and test various molecular markers including NF-kB signaling, inflammasome, ROS and fibrosis. The results are solid and informative. Some minor concerns were provided below:

1. the liver consists of hepatocyte, stellate cell, and inflammatory cells. Through which cell type do the authors think caffeine inhibits NLRP3 inflammasome?

2. some background information about IL-17’s role in inflammasome and NF-kB signaling should be added to manuscript to support Fig. 4G.

3. Hepatocyte are major lipid producers, and stellate cells and liver infiltrated inflammatory cells are major contributors of inflammation in the NASH development. According to the results, caffeine doesn’t show any improvement in hepatocyte steatosis (Fig. 2) but seems protect against NASH via its anti-inflammation effect. Some discussion about caffeine’s cell-specific effect in the liver should be discussed.

Author Response

Reviewer 1

The authors test caffeine effect in a NASH model, and test various molecular markers including NF-kB signaling, inflammasome, ROS and fibrosis. The results are solid and informative. Some minor concerns were provided below:

  1. the liver consists of hepatocyte, stellate cell, and inflammatory cells. Through which cell type do the authors think caffeine inhibits NLRP3 inflammasome?

Answer:

Thank you for your comment.

NLRP3 inflammasome is most prominently expressed in Kupffer cells and liver sinusoidal endothelial cells and moderately expressed in periportal myofibroblasts, hepatic stellate cells, and hepatocytes [10,45]; therefore, caffeine NLRP3 inhibition may principally occur in Kupffer cells; however, the effect of caffeine on other cell types cannot be discarded with the present data.

Please see the new version of the manuscript on Page 12, Lines 369 to 372.

  1. some background information about IL-17’s role in inflammasome and NF-kB signaling should be added to manuscript to support Fig. 4G.

Answer:

Thank you for your comment.

The NF-κB signaling pathway promotes the synthesis of NLRP3, pro-IL-1β, and other proinflammatory cytokines. Activated NLRP3 inflammasome and IL-1β promote the production of the proinflammatory and profibrogenic IL-17, which contributes to ECM exacerbated deposition within the hepatic parenchyma [10,45,47]. Consequently, inflammasomes are now major drug targets for many chronic inflammatory diseases, such as NASH.

Please see the new version of the manuscript on Page 13, Lines 391 to 393.

  1. Hepatocyte are major lipid producers, and stellate cells and liver infiltrated inflammatory cells are major contributors of inflammation in the NASH development. According to the results, caffeine doesn’t show any improvement in hepatocyte steatosis (Fig. 2) but seems protect against NASH via its anti-inflammation effect. Some discussion about caffeine’s cell-specific effect in the liver should be discussed.

Answer:

Thank you for your comment.

NLRP3 inflammasome is most prominently expressed in Kupffer cells and liver sinusoidal endothelial cells and moderately expressed in periportal myofibroblasts, hepatic stellate cells, and hepatocytes [10,45]. Therefore, caffeine NLRP3 inhibition may principally occur in Kupffer cells. On the other hand, free fatty adds that originate from lipolysis of triglyceride in adipose tissue are delivered through the blood to the liver. The other major contributor to the free fatty acid flux through the liver is de novo lipid synthesis, the process by which hepatocytes convert excess carbohydrates, especially fructose, to fatty acids. The two major fates of fatty acids in hepatocytes are mitochondrial beta-oxidation and re-esterification to form triglyceride. Triglyceride can be exported into the blood as very low-density lipoprotein or stored in lipid droplets. Lipid droplet triglyceride undergoes regulated lipolysis to release fatty acids back into the hepatocyte-free fatty acid pool [11,46]. Our results show no significant effect of caffeine on the fatty liver but an important anti-inflammatory effect; therefore, it seems that caffeine exhibits a cell-specific effect, acting on proinflammatory cells but not on hepatocytes. However, the effect of caffeine on other cell types cannot be completely discarded with the present data. 

Please see the new version of the manuscript on Pages 12 and 13, Line 369 to 384.

References:

  1. Alegre, F.; Pelegrin, P.; Feldstein, A.E. Inflammasomes in liver fibrosis. Semin Liver Dis. 2017, 37(2):119-127. doi:10.1055/s-0037-1601350
  2. Friedman, S.L.; Neuschwander-Tetri, B.A.; Rinella, M.; et al. Mechanisms of NAFLD development and therapeutic strategies. Nat Med. 2018, 24(7):908-922. doi:10.1038/s41591-018-0104-9
  3. Boaru, S.G.; Borkham-Kamphorst, E.; Tihaa, L., Haas, U.; Weiskirchen, R. Expression analysis of inflammasomes in experimental models of inflammatory and fibrotic liver disease. J Inflamm (Lond). 2012, 9(1):49. doi:10.1186/1476-9255-9-49

46.- Ipsen, D.H.; Lykkesfeldt, J.; Tveden-Nyborg, P. Molecular mechanisms of hepatic lipid accumulation in non-alcoholic fatty liver disease. Cell Mol Life Sci. 2018, 75(18):3313-3327. doi: 10.1007/s00018-018-2860-6

  1. Mills, K.H.; Dungan, L.S.; Jones, S.A.; Harris, J. The role of inflammasome-derived IL-1 in driving IL-17 responses. J Leukoc Biol. 2013, 93(4):489-497. doi:10.1189/jlb.1012543

Reviewer 2 Report

This study aimed to evaluate the effect of caffeine on the NLRP3 inflammasome signaling pathway in a rat model of NASH. This is an interesting study, because caffeine as a life style modification is regarded as one of disease modifier in various areas.

There are minor comments to be addressed.

1) What about intrahepatic steatosis after adding caffeine?

2) Caffeine might affect various extra-hepatic parameters associated with course of NAFLD.  They should be analyzed, if eligible (e.g. blood lipid profile, gut microbiome,,,)

Author Response

Reviewer 2

This study aimed to evaluate the effect of caffeine on the NLRP3 inflammasome signaling pathway in a rat model of NASH. This is an interesting study, because caffeine as a life style modification is regarded as one of disease modifier in various areas.

There are minor comments to be addressed.

1) What about intrahepatic steatosis after adding caffeine?

Answer:

Thank you for your comment.

We also found that caffeine does not prevent liver lipid accumulation, which suggests that its hepatoprotective effects are not due to its antisteatotic capability. By contrast, other re-searchers have reported that caffeine exerts antisteatotic effects. These opposing effects may be ascribed to the fact that the effect of caffeine was determined in combination with other compounds [17,18]. According to our results, caffeine does not show any improvement in hepatocyte steatosis but seems to protect against NASH via its anti-inflammation effect.

Please see the new version of the manuscript on Page 12, Lines 345 to 347.

2) Caffeine might affect various extra-hepatic parameters associated with course of NAFLD.  They should be analyzed, if eligible (e.g. blood lipid profile, gut microbiome,,,)

Answer:

Thank you for your suggestion.

It is known that the development of NAFLD and NASH is closely related to metabolic syndrome which includes disorders such as high blood pressure, high blood sugar, excess body fat around the waist, abnormal blood lipid levels, and alteration in the gut microbiome. In this sense, previous investigations have shown that no associations were found between caffeine concentrations with total cholesterol and low-density lipoprotein levels either in caffeine-drug users or nonusers [24], other report shows that caffeine does not interfere with the lipid profile in cyclists [25]. Concerning the intestinal microbiome, a study showed that caffeine is directly associated with changes only in some intestinal microbiota groups such as the Bacteroides group [26]. Another study showed that higher caffeine consumption was associated with increased richness and evenness of the mucosa-associated gut microbiota, and higher relative abundance of anti-inflammatory bacteria, such as Faecalibacterium and Roseburia, and lower levels of potentially harmful Erysipelatoclostridium [27]. Regarding other metabolic disorders, it is known that in persons who have not previously consumed caffeine, caffeine intake raises blood pressure in the short term, affect tolerance develops within a week but may be incomplete in some person results in a modest increase in systolic and diastolic blood pressure. Experimental studies in humans do not show an association between caffeine intake and atrial fibrillation. Other findings indicate that the consumption of caffeinated coffee is not associated with an increased risk of cardiovascular events in the general population or among persons with a history of hypertension, diabetes, or cardiovascular diseases. Metabolic studies suggest that caffeine may improve energy balance by reducing appetite and increasing the basal metabolic rate and food-induced thermogenesis. Limited evidence from randomized trials also supports a modest beneficial effect of caffeine intake on body fatness. Consumption of caffeinated coffee for up to 6 months does not affect insulin resistance [28].

Please see the new version of the manuscript on Pages 11 and 12, Lines 316 to 339.

References

  1. Murosaki, S.; Lee, T.R.; Muroyama, K.; et al. A combination of caffeine, arginine, soy isoflavones, and L-carnitine enhances both lipolysis and fatty acid oxidation in 3T3-L1 and HepG2 cells in vitro and in KK mice in vivo. J Nutr. 2007, 137(10):2252-2257. doi:10.1093/jn/137.10.2252
  2. Sugiura, C.; Nishimatsu, S.; Moriyama, T.; et al. Catechins and caffeine inhibit fat accumulation in mice through the improvement of hepatic lipid metabolism. J Obes. 2012, 2012:520510. doi:10.1155/2012/520510
  3. Du, Y.; Melchert, H.U.; Knopf, H.; Braemer-Hauth, M.; Gerding, B.; Pabel, E. Association of serum caffeine concentrations with blood lipids in caffeine-drug users and nonusers - results of German National Health Surveys from 1984 to 1999. Eur J Epidemiol. 2005, 20(4):311-6. doi: 10.1007/s10654-004-7536-x
  4. Marangon, A.F.C.; Helou, T.; Gonzalez, D.V. Effect of caffeine on lipid profile in ciclism practitioners. J Int Soc Sports Nutr. 2012, 9(Suppl 1):P20. doi: 10.1186/1550-2783-9-S1-P20
  5. González, S.; Salazar, N.; Ruiz-Saavedra, S.; Gómez-Martín, M.; de Los Reyes-Gavilán C.G.; Gueimonde, M. Long-term coffee consumption is associated with fecal microbial composition in humans. Nutrients. 2020, 12(5):1287. doi: 10.3390/nu12051287
  6. Gurwara, S., Dai, A., Ajami, N., El-Serag, H.B., Graham, D.Y., Jiao, L. Caffeine consumption and the colonic mucosa-associated gut microbiota. Off J Am Coll Gastroenterol. 2019, 114, S119–S120. doi: 10.14309/01.ajg.0000590316.43252.64
  7. van Dam, R.M.; Hu, F.B.; Willett, W.C. Coffee, caffeine, and health. N Engl J Med. 2020, 383(4):369-378. doi: 10.1056/NEJMra1816604

Round 2

Reviewer 2 Report

Authors addressed raised issues appropriately.